# Primary Definitive Treatment versus Ureteric Stenting in the Management of Acute Ureteric Colic: A Cost-Effectiveness Analysis

**DOI:** 10.3390/jpm12111773

**Published:** 2022-10-27

**Authors:** Radha Sehgal, Yasmin Abu-Ghanem, Christina Fontaine, Luke Forster, Anuj Goyal, Darrell Allen, Rajesh Kucheria, Paras Singh, Gidon Ellis, Leye Ajayi

**Affiliations:** Department of Urology, Royal Free London NHS Foundation Trust, London NW3 2QG, UK

**Keywords:** ureteric stones, temporising measures, ureteric stent, definitive treatment, ureteroscopy, extracorporeal shock wave lithotripsy, cost

## Abstract

Objectives: To analyze the differences in cost-effectiveness between primary ureteroscopy and ureteric stenting in patients with ureteric calculi in the emergency setting. Patients and Methods: Patients requiring emergency intervention for a ureteric calculus at a tertiary centre were analysed between January and December 2019. The total secondary care cost included the cost of the procedure, inpatient hospital bed days, emergency department (A&E) reattendances, ancillary procedures and any secondary definitive procedure. Results: A total of 244 patients were included. Patients underwent ureteric stenting (62.3%) or primary treatment (37.7%), including primary ureteroscopy (URS) (34%) and shock wave lithotripsy (SWL) (3.6%). The total secondary care cost was more significant in the ureteric stenting group (GBP 4485.42 vs. GBP 3536.83; *p* = 0.65), though not statistically significant. While mean procedural costs for primary treatment were significantly higher (GBP 2605.27 vs. GBP 1729.00; *p* < 0.001), costs in addition to the procedure itself were significantly lower (GBP 931.57 vs. GBP 2742.35; *p* < 0.001) for primary treatment compared to ureteric stenting. Those undergoing ureteric stenting had a significantly higher A&E reattendance rate compared with primary treatment (25.7% vs. 10.9%, *p* = 0.02) and a significantly greater cost per patient related to revisits to A&E (GBP 61.05 vs. GBP 20.87; *p* < 0.001). Conclusion: Primary definitive treatment for patients with acute ureteric colic, although associated with higher procedural costs than ureteric stenting, infers a significant reduction in additional expenses, notably related to fewer A&E attendances. This is particularly relevant in the COVID-19 era, where it is crucial to avoid unnecessary attendances to A&E and reduce the backlog of delayed definitive procedures. Primary treatment should be considered concordance with clinical judgement and factors such as patient preference, equipment availability and operator experience.

## 1. Introduction

The growing prevalence of nephrolithiasis carries a significant financial burden, making cost-reduction management of these patients imperative. In the UK, hospital episodes related to upper urinary tract stones have increased by 63% from 2000–2010, with a further increase of 4.4% from 2010–2015, accounting for 6000 hospital admissions annually [1,2,3].

Although most cases of renal colic presenting to the accident and emergency department (A&E) will not require surgical intervention due to spontaneous stone passage, up to 20% of patients will ultimately undergo surgical intervention [4].

Until such a decision is taken, these patients may suffer persistent colic pain, which may lead to frequent visits to the A&E, recurrent hospital admissions and loss of working days, all of which are associated with a significant economic burden. Internationally, expenditure associated with a stone disease is on the rise; in the USA alone, the annual costs are estimated to be more than USD 5 billion [5].

Until recently, the conventional management of acute ureteric colic included either retrograde ureteric stenting or percutaneous nephrostomy, followed by delayed definitive treatment of the stone. However, in the last few years, emergency stone clearance by ureteroscopy (URS) or shock wave lithotripsy (SWL) has emerged as reasonable first-line treatment options for ureteric stones to avoid long-term stenting [6,7,8,9,10].

This practice is now recommended by the National Institute for Health and Care Excellence (NICE) guidelines, which recommend offering definitive treatment (defined as primary URS or SWL) to patients with ureteric stones and renal colic within 48 h of diagnosis in patients who are unlikely to pass their stones or with ongoing pain [11]. In support of the NICE recommendation, a randomised prospective trial by Guercio et al. revealed higher stone-free rates and significantly fewer stents inserted in the early versus delayed treatment group [12]. Similarly, Kumar et al. demonstrated significantly lower mean time to stone clearance, fewer sessions and fewer ancillary procedures in patients undergoing SWL within 48 h of presentation [13]. In addition to the superior clinical outcomes, NICE predict an annual saving of GBP 11 million due to avoided stents alone [11].

However, despite these promising studies, there have been limited studies conducted in countries with public healthcare systems such as the UK, particularly with a view to cost-analysis. 

The current study aimed to review the cost-effectiveness of primary definitive treatment of ureteric stones (defined as early URS or SWL) compared to emergency decompression with ureteric stenting with subsequent delayed definitive treatment. 

## 2. Materials and Methods

### 2.1. Study Design and Participants

Following institutional approval, we performed a retrospective analysis of all patients requiring emergency intervention for computed tomography (CT) confirmed ureteric calculus at our institution. All patients underwent either ureteric stenting, primary URS or SWL between January to December 2019. Indications for intervention were: pain that was non-responsive to anti-inflammatory agents and acute renal failure, defined as an increase in serum creatinine above the patient’s baseline. The responsible surgeon decided the choice of intervention. This study did not include patients managed expectantly or that did not require urgent intervention. 

Patients with an infected obstructed system were excluded from the analysis.

Data were collected by reviewing electronic records, including medical notes, operation notes, picture archiving and communication system (PACS) and discharge summaries. Demographic data recorded were: age, gender, time of admission, stone size, location and laterality. “In hours (IH)” was defined as patients admitted between the hours of 08:00 and 17:00 and “out of hours (OOH)” between the hours of 17:00 and 08:00. Primary outcomes compared were: operative cost, costs arising in addition to the procedure itself and overall secondary care cost. Secondary outcomes were: treatment success rate, time to intervention, length of postoperative stay (LOS) and the number of revisits to A&E.

Successful treatment was defined as a primary treatment (ureteroscopy or SWL) where stone removal was achieved, confirmed by documentation in the operation notes or on follow-up CT. Retrograde ureteric stenting was deemed successful, where satisfactory ureteric stent insertion was achieved.

### 2.2. Surgical Techniques

Emergency primary URS was undertaken with the intent to provide definitive treatment. All cases were performed under general anaesthesia with a semi-rigid ureteroscope. Stones were either fragmented using holmium: YAG laser if required or removed by an endoscopic basket. A flexible ureterorenoscopy was available where necessary. The decision to insert a ureteric stent and the duration this remained in situ was at the discretion of the surgeon.

SWL was performed under local anaesthetic by the same dedicated radiographer in all cases using an on-site lithotripter (Storz Medical Modulith SLX-F2, Tägerwilen, Switzerland). The number of shocks delivered varied depending on stone size and density up to a maximum of 3000 pulses. Maximum shockwave energy and speed delivered were 7 J and 4 Hz, respectively. Patients undergoing SWL did not have a ureteric stent inserted before or after the procedure.

Retrograde ureteric stenting was performed under general anaesthesia. All stents were inserted via rigid cystoscopy using fluoroscopic guidance. Stent size was selected on an individual patient basis. On discharge, patients were scheduled for a definitive elective procedure. Ureteric stents were removed during definitive elective surgery via a string or in clinic by flexible cystoscopy under local anaesthetic if the patient’s stone had passed. Unless contra-indicated, patients are given an alpha-blocker and anticholinergic to pre-emptively treat stent symptoms. 

### 2.3. Financial Data

Our institution’s costing data was collated from the clinical coding department, considering the greater costing tariffs for emergency versus elective operations. Tariffs are consistent regardless of the time of operation, and costs of primary URS assume peri-operative stenting. Unit costs for inpatient bed days and emergency department attendances were taken from NHS reference costs 2017/18 [14].

Costs were divided into procedural costs, additional costs and the total secondary care cost. The Procedural Cost included the cost of the emergency procedure alone (emergency primary URS, SWL sessions or retrograde ureteric stenting) according to our institution. The additional costs were costs arising other than those of the procedure itself, including:Inpatient hospital bed days related to post-procedure LOSA&E revisits within three months (inpatient admissions were negligible and therefore excluded from analysis)Ancillary procedures such as nephrostomyDefinitive elective ureteroscopy if applicable.

The total secondary care cost was a combination of procedural and additional costs.

### 2.4. Statistical Analysis

Statistical analysis was performed using Statistical Package for Social Sciences (SPSS, Version 22.0, Chicago, IL, USA). Data are presented as median (interquartile range, IQR) or number (per cent) unless otherwise specified. The student’s *t*-test and the Mann–Whitney U-test were used to analyse continuous variables, and the Chi-square test was used for categorical variables. A *p*-value of <0.05 was considered statistically significant.

## 3. Results

The current study included 257 patients who required emergency intervention for ureteric calculi. Among these, 13 patients presented with an infected obstructed urinary system and thus were excluded from analysis. A total of 244 patients were included in the final analysis. Retrograde ureteric stenting was performed in 152 patients (62.3%) and primary treatment in 92 patients (37.7%), of which: 83 patients (34.0%) underwent emergency URS and 9 patients (3.6%) had SWL (Figure 1).

### 3.1. Patient Demographics 

Patients’ demographic profiles and stone characteristics are summarised in Table 1. Age, gender, time of admission, stone size and laterality were comparable between the ureteric stenting and primary treatment groups. Patients undergoing primary treatment were likelier to have a distally located stone (*p* = 0.001).

### 3.2. Treatment Outcomes 

Overall, the success rate for primary stone treatment was 83.7%. For most patients undergoing SWL this was achieved in 1 session (55%), with the remainder requiring two sessions (45%). Successful primary URS was performed in 68 patients (82%). A ureteric stent was inserted at the end of the procedure in 78 cases (94.0%), of which 10 (12%) were left in situ with strings attached. The mean time to ureteric stent removal was 30 days. 

The success rate for retrograde stenting was 98%. The mean time to the secondary definitive procedure was 82 days.

Primary URS was unsuccessful in 15 cases (18%) compared to ureteric stenting, which failed in 3 cases (2.0%). Reasons for treatment failure and subsequent treatment are summarised in Figure 2. On further analysis of these patients, stone size and position were not significant risk factors for treatment failure.

Mean time to ureteric stenting was longer than primary treatment, though not significant (30.7 ± 30.6 and 25.2 ± 22.1 h, respectively) (*p* = 0.16). Over 80% underwent their procedure within 48 h of diagnosis; 126 patients (82.9%) in the ureteric stenting group and 76 patients (82.6%) in the primary treatment group. 

Mean postoperative LOS was longer in the ureteric stenting group (3.1 days) compared with the primary treatment group (2.6 days). However, this was not statistically significant (*p* = 0.10). 

The number of reattendances to A&E is shown in Table 2. Reattendance was significantly lower in patients who underwent primary treatment than ureteric stenting (*p* = 0.02). Three patients were readmitted in the ureteric stenting group due to urinary infection. All other reattendances were due to stent-related symptoms (pain/dysuria/frequency).

### 3.3. Cost Analysis 

Table 3 summarises the average procedural costs at our institution as per best practice tariffs. 

The mean procedural cost was GBP 1729.00 for ureteric stenting and GBP 2605.27 for the primary treatment (mean difference of GBP 876.27; *p* < 0.001). 

The mean additional costs incurred per patient (costs other than the surgical procedure itself) were GBP 2742.35 for ureteric stenting and GBP 931.57 for primary treatment (mean difference GBP 1810.79; *p* < 0.001). When excluding patients where no additional costs were incurred in both the ureteric stenting group (*n* = 3) and primary treatment group (*n* = 17), additional costs for ureteric stenting were GBP 2792.57, and primary treatment were GBP 1142.72 (mean difference GBP 1654.85; *p* < 0.001).

The total secondary care cost up to and including the patient’s definitive stone treatment was higher in the ureteric stenting group (mean cost GBP 4485.42) than in the primary treatment group (mean cost GBP 3536.83), but this was not statistically significant (mean difference GBP 948.59; *p* = 0.65). A summary of cost outcomes is displayed in Figure 3. 

On sub-analysis of additional costs, the mean cost per patient related to LOS was GBP 1056.21 in those who underwent ureteric stenting compared to GBP 891.33 in patients receiving primary treatment (*p* = 0.99). The cost specifically related to reattendances to A&E totalled GBP 9280 in the ureteric stenting group compared with GBP 1920 in the primary treatment group, amounting to a significantly higher cost per patient (GBP 61.05 vs. GBP 20.87; *p* < 0.001). 

## 4. Discussion

The current study included a cohort of 244 patients requiring emergency intervention for acute ureteric colic during a 1-year period at our centre, which has the facility to offer primary ureteroscopy and acute SWL as well as emergency ureteric stenting. To our knowledge, this is the most extensive cost-effectiveness study to include all three modes of treatment. 

Our data demonstrated that primary treatment was more likely to be performed in those with distal ureteric stones, regardless of the size of the stone. This concurs with Osorio et al., who, in their retrospective analysis of patients undergoing emergency ureteroscopy, 90.3% of stones were located in the distal ureter [6]. A potential explanation may be that a clinician would opt for ureteric stenting in the case of a proximal stone due to perceived difficulty in retrieving the stone, time pressures and availability of subspecialist staff. Interestingly, we found no other variable in the current cohort determined which patients underwent primary treatment compared with ureteric stenting. Discussion of the factors that influence the decision to perform URS, SWL or ureteric stenting is beyond the scope of this paper. 

As per NICE guidance, 82.9% of patients underwent ureteric stenting, and 82.6% received primary treatment within 48 h. This practice has proven cost-effective and improved time to stone-free status and fewer ureteric stents [11].

Our treatment success rate for ureteric stenting was 98%, and primary ureteroscopy was 83.7%. These rates are comparable with other studies where emergency URS was performed in a clinical setting [7,8].

There was a trend for a longer mean postoperative LOS of 3.1 days in those who underwent ureteric stenting compared to 2.6 days in the primary treatment group. Although these procedures can be considered day-case procedures electively, in the acute setting, discharge may be delayed whilst parameters such as abnormal renal function normalise. 

We also found a significantly higher reattendance rate to A&E in the ureteric stenting group. Over one-quarter of stented patients reattended the emergency department during our study period. Most of these presentations were due to symptoms secondary to the ureteric stent, such as acute pain, urinary frequency and dysuria. Ureteric stents are associated with significant morbidity and have been shown to harm quality of life. Joshi et al. used a validated questionnaire to assess stent-related symptoms and demonstrated that over 80% of patients experienced severe pain to impact daily activities [15]. Similarly, Leibovici et al. found that 32% of their cohort were still absent from work after one month [16]. These side effects are a substantial contributing factor in increased LOS and reattendance to hospitals, leading to increased healthcare expenditure and negative economic impact.

The current cost analysis demonstrates that the total secondary care cost tends to be higher for ureteric stenting than primary treatment, though not statistically significant. However, a separate analysis of procedural costs and additional costs showed the additional costs for ureteric stenting to be statistically significantly higher than primary treatment. The procedural cost was significantly higher for primary treatment than ureteric stenting, accounted for by the highest individual costing tariff of emergency ureteroscopy. Similar findings have been shown by the few published studies on the cost-effectiveness of primary treatment vs. stenting. Darrad et al. evaluated the cost-effectiveness of primary URS vs. ureteric stenting in a smaller cohort of 50 patients. They demonstrated that primary ureteroscopy was significantly more cost-effective than ureteric stenting for estimated operative and overall secondary care costs [17]. Similarly, another cost-effectiveness study has shown that URS is the most cost-effective modality after conservative measures have failed [18].

On further analysis, we found a significantly higher cost related to A&E reattendances for patients who underwent ureteric stenting than those who underwent primary treatment (GBP 61.05 vs. GBP 20.87). This correlates with the increased revisits to A&E secondary to stent-related symptoms. Patients with a primary procedure are more likely to be discharged stent-free, with a stent on a string or undergo earlier stent removal than those awaiting their secondary definitive procedure. Particularly in the era of the COVID-19 pandemic, prevention of avoidable A&E reattendance is an essential consideration when choosing between treatment modalities due to the cost implications as well as the burden on the emergency department. 

Limitations of this study include the small sample size of those undergoing SWL within the primary treatment group. Most patients underwent primary URS, which is most established at our institution, and our acute SWL is relatively novel. Nevertheless, it is essential that, if available, SWL is reflected in the primary treatment of acute ureteric colic. Secondly, due to the study’s retrospective design, surgeon preference and experience may have influenced the decision-making, including the choice of treatment. However, given that focus of this study is cost-analysis of the different treatment modalities versus the comparison of clinical outcomes of each intervention, we were able to overcome this limitation.

Moreover, although the analysis incorporated a breadth of secondary care costs, we did not consider the financial impact of reduced work capacity as this was beyond the scope of our study.

Moreover, given the retrospective nature of the study, inherited bias is inevitable. Furthermore, we were not able to gather precise additional data including costs for medication of stent related symptom. We therefor believe that further prospective study may allow for more accurate cost evaluation of all additional costs. 

## 5. Conclusions

This study highlights that whilst emergency ureteric stenting in patients with acute ureteric colic has a lower procedural cost than primary treatment, it is associated with significantly higher additional costs when taking into account inpatient stay, A&E visits, ancillary procedures and definitive surgery. Clear savings were found for primary treatment when avoiding reattendances to the A&E department. This is pertinent in the COVID-19 pandemic, where reducing unnecessary A&E attendances and the backlog of delayed definitive procedures is crucial. We conclude that in agreement with current NICE guidance, primary URS and SWL in the acute setting should be considered to avoid reattendances to A&E, reduce LOS and confer a potential overall secondary care cost benefit. In addition to cost-effectiveness, the choice of procedure should be based on clinical judgement and consider factors such as patient preference, equipment availability and operator experience. 

## Figures and Tables

**Figure 1 jpm-12-01773-f001:**
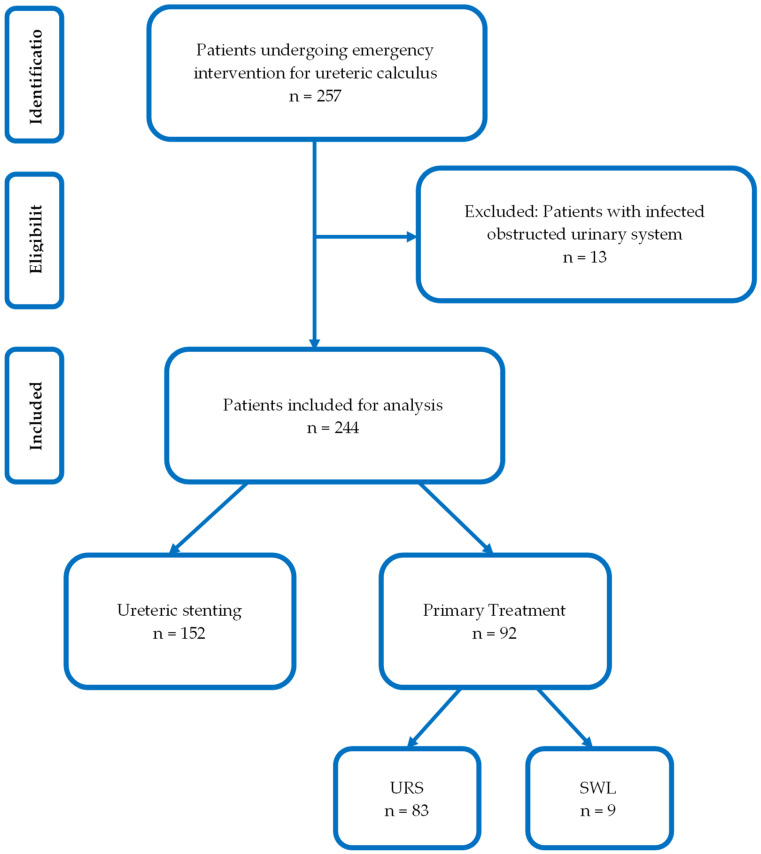
Flowchart demonstrating inclusion and exclusion criteria for the present study.

**Figure 2 jpm-12-01773-f002:**
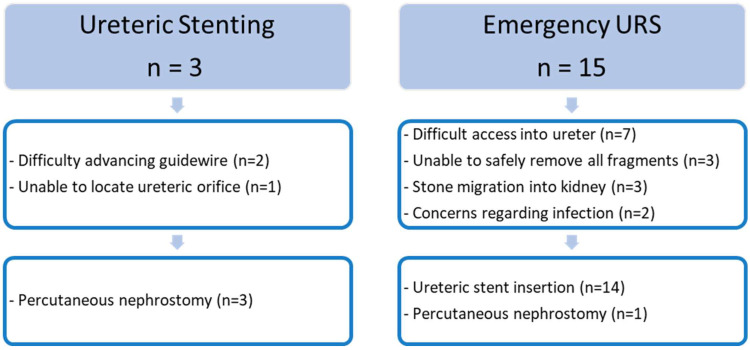
Reasons for treatment failure and subsequent treatment.

**Figure 3 jpm-12-01773-f003:**
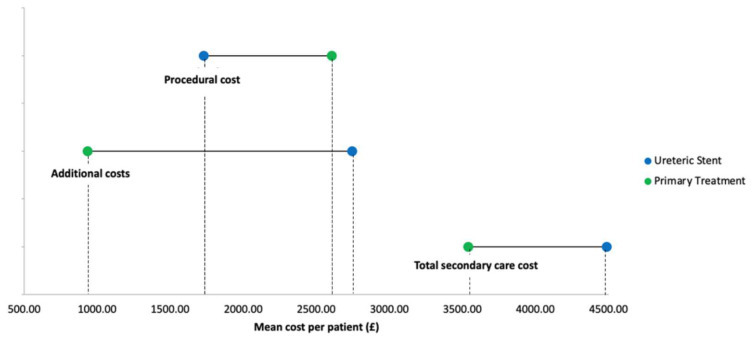
Summary of cost outcomes.

**Table 1 jpm-12-01773-t001:** Demographic variables.

Variable	Ureteric Stenting (*n* = 152)	Primary Treatment (*n* = 92)	*p* Value
Mean age (years)	52.1	50.6	0.73
Sex (M/F)	97/55	64/28	0.36
Time of admission (IH/OOH)	57/95	39/53	0.45
Mean Stone size (mm)	7.8	6.9	0.14
Stone location (Proximal/Distal)	93/59	30/62	<0.001
Stone laterality (Left/Right)	78/74	42/50	0.39

M-male; F-female; IH-in hours; OOH-out of hours.

**Table 2 jpm-12-01773-t002:** Number of Reattendances to Emergency Rooms.

Reattendances to A&E		0	1	≥2
Ureteric Stenting (*n* = 152)	*n*	113	28	11
	%	74.3%	18.4%	7.2%
Primary Treatment (*n* = 92)	*n*	82	8	2
	%	89.1%	8.7%	2.2%
Total	*n*	195	36	13
	%	79.9%	14.8%	5.3%

A&E-Accident and Emergency.

**Table 3 jpm-12-01773-t003:** Overall cost per procedure/visit.

Activity	Unit Cost
Emergency ureteric stent	GBP 1729.00
Emergency URS	GBP 2829.39
SWL (single session)	GBP 542.00
Elective URS	GBP 2138.65
Percutaneous nephrostomy	GBP 1782.00
Inpatient bed days	GBP 346.00
A&E attendance	GBP 160.00

A&E-Accident and Emergency; URS- ureteroscopy; SWL-Shock Wave Lithotripsy.

## Data Availability

Not applicable.

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
