# Peer review of "Primary Definitive Treatment versus Ureteric Stenting in the Management of Acute Ureteric Colic: A Cost-Effectiveness Analysis"

_jpm, 2022, doi:10.3390/jpm12111773_

Round 1

Reviewer 1 Report

Dear Editor,

With great interest I read the article by Sehgal et al regarding cost-effectiveness of direct treatment vs stenting in acute renal colic. The topic they describe is relevant for daily clinic. Although we are mainly focused on treatment effect, we sometimes are unaware of the actual costs we make.

I have some small comments that need to be addressed. Overall I was very pleasantly surprised by this work.

-          The set-up of the study (retrospective) can cause some bias in this study. Nevertheless, I think this factor is relatively small. A prospective study will result in better tracking of actual costs.  

-          In the cost-analysis I miss: costs for medication of stent related symptom. Costs for removal of ureteric stents at the outpatient clinic.

Author Response

The authors are grateful for the reviewers’ comments. In regard to the retrospective nature of the study as well as the additional costs, we agree that a prospective study will allow us to draw better conclusions with less bias and more information.  A comment was added to the discussion.

Reviewer 2 Report

Great review of an age old question. the inclusion of ESWL is interesting but has limited applicability as most institutions cannot offer this in real time for acute stones. Otherwise, good review and inclusion of NICE guidelines.

Author Response

Many thanks for your comments.